# The African Psyllid *Trioza erytreae* Del Guercio (1918) Is Very Sensitive to Low Relative Humidity and High Temperatures

**DOI:** 10.3390/insects15010062

**Published:** 2024-01-16

**Authors:** Rosa Pérez-Otero, Raquel Pérez-Turco, Joana Neto, Alberto Fereres

**Affiliations:** 1Estación Fitopatolóxica Areeiro, Deputación de Pontevedra, Subida a la Robleda, s/n, 36153 Pontevedra, Spain; rosa.perez@depo.es (R.P.-O.); raquelpeturco@hotmail.com (R.P.-T.); 2GreenUPorto—Sustainable Agrifood Production Research Centre/Inov4Agro, Faculty of Sciences, University of Porto, Rua da Agrária 747, 4485-646 Vairão, Portugal; joana.isa.neto@gmail.com; 3Instituto de Ciencias Agrarias, ICA-CSIC, Calle Serrano 115 dpdo, 28006 Madrid, Spain

**Keywords:** development, mortality, fecundity, fertility, African citrus psylla, climate change

## Abstract

**Simple Summary:**

Our study shows that the African citrus psyllid, a vector of Huanglongbing disease, prefers to develop at mild temperatures and is not well adapted to dry regions, where relative humidity is around or below 40%. Its life cycle is not completed or is severely delayed at temperatures above 30 °C and below 10 °C. Eggs are unable to hatch at constant temperatures below 8 °C or above 33 °C. Thus, *T. erytreae* establishment and spread will be maximum in regions with a temperate and humid climate, being rare in regions where dry weather conditions predominate.

**Abstract:**

The African citrus psyllid, *Trioza erytreae,* is one of the two vectors of Huanglongbing, the most serious citrus disease worldwide. The first detection of *T. erytreae* in the European mainland was on the northwest of the Iberian Peninsula in 2014. Since then, the pest has spread throughout northern Spain (Galicia, Asturias, Cantabria, País Vasco) and along the western Atlantic coast of Portugal (from the Douro e Minho region to the Algarve). We conducted a series of laboratory experiments on lemon plants at different temperatures (from 8 to 34 °C) and humidity conditions (from 40 to 90%) to find out the influence of extreme temperatures and relative humidities (RHs) on the mortality, development and reproduction of *T. erytreae*. Our results show that temperatures above 30 °C and below 10 °C are very detrimental for nymphal development and nymphs were unable to reach the adult stage. Furthermore, eggs were unable to hatch under temperatures above 33 °C and below 8 °C. Adult mortality was highest at 34 °C and killed more than 50% of the population. We also found that relative humidity is crucial for the development and survival of *T. erytreae*. Nymphs were unable to reach the adult stage at an RH of 90% and 40%. Also, fecundity was significantly reduced at 90 and 40% RH, and fertility was lowest at 40% RH. Nymphal mortality was highest at an RH of 40%, which was the most detrimental humidity among all tested for the survival and development of *T. erytreae*. Our work concludes that *T. erytreae* establishment and spread will be maximum in regions with a temperate and humid climate, being rare in regions where dry and hot weather conditions predominate.

## 1. Introduction

Globalization and the ease with which humans can travel long distances, along with climate change, are the triggers for the increase in emerging agricultural diseases worldwide [1], which pose a serious threat to agricultural production. Among them are diseases caused by bacteria associated with the xylem and phloem, which are mainly transmitted by insects of the order Hemiptera. The African citrus psyllid, *Trioza erytreae* (Del Guercio, 1918) (Hemiptera: Triozidae), is the vector of the bacterium *Candidatus* Liberibacter africanus or *C*Lafr [2] and is known to efficiently transmit another, even more aggressive, species of the bacteria in nature: *Candidatus* Liberibacter asiaticus or *C*Las [3] Both *C*Lafr and *C*Las are the causative agents of the disease known as Huanglongbing (HLB) or greening, currently considered the most destructive pathology affecting citrus trees [4], as evidenced by the fact that it has already caused the death of millions of trees in different citrus-growing regions such as Florida or Brazil [5].

The damage caused by HLB includes the production of asymmetrical fruits, poorly colored with aborted seeds of little commercial value due to their reduced size and quality [6]. In areas where the disease is epidemic, trees show decay and die in a few years [7]. Currently, HLB is present in almost all citrus-producing regions of the world, except in the Mediterranean area.

Such is their potential significance that *C*Las and *C*Lafr are not only quarantine pests according to Implementing Regulation (EU) 2019/2072 (which builds on Regulation (EU) 2016/2031 regarding protective measures against plant pests) but also priority pests under Regulation (EU) 2019/1702. To date, HLB has not been detected in Europe but both psyllid vectors of HLB are already present in the Mediterranean, with *T. erytreae* in Portugal and Spain and *Diaphorina citri* in Israel and Cyprus [8].

*Trioza erytreae* was first described from Eritrea, then part of Ethiopia, by Del Guercio (1918), although its presence had already been reported in South Africa in 1897 (as “citrus psylla”) [9]. Since then, *T. erytreae* has appeared in other sub-Saharan African countries, as well as in areas of Saudi Arabia and Yemen in Asia [10]. *Trioza erytreae*, also a quarantine pest in Europe under Implementing Regulation (EU) 2019/2072, was first detected in Madeira in 1994 and then in the Canary Islands in 2002 [11]. In 2014, it was found in the European mainland, in the northwest of the Iberian Peninsula [12]. Since then, it has spread to other Spanish regions along the northern coast (Cantabric Sea) and to the Atlantic coast of western Portugal as far as the Algarve [13].

*Trioza erytreae* affects young shoots, where it deforms the leaves markedly, can cause chlorosis, and can facilitate the development of fungi such as *Capnodium* sp. on the honeydew secreted by the nymphs, attracting ants that disturb the protective action of natural enemies [11]. However, in adult trees, *T. erytreae* do not cause severe direct damage; Tamesse and Messi (2002) [14] demonstrated that, in the absence of control measures, losses in nurseries could reach 91%, and the growth of seedlings could be severely affected. If the bacteria causing HLB become established in Europe, the damage to citrus could be dramatic, as *T. erytreae* can act as an efficient vector of the most aggressive bacterial strain, *C*Las.

The current spread of the pest already poses an immediate threat to citrus production in the southern regions of the Iberian Peninsula. In this context, different research projects are being developed that focus on controlling *T. erytreae* and preventing the introduction of HLB from various approaches. One of the outcomes of these projects focuses on the use of cultural control strategies including the use of kaolin to interfere with adults landing on lemon orchards [15]. Other projects have successfully facilitated the establishment of a classical biological control program based on the introduction of *Tamarixia dryi* (Waterston) in Spain and Portugal [16], obtaining excellent results.

Various studies on the bioecology of *T. erytreae* can be found in the literature, referring to the relationship of certain climatological conditions that limit the different phases of its life cycle ([17,18,19,20,21]). Many of them are based on studies or observations carried out decades ago under field or semi-field conditions in the Southeast of the African continent.

Although all these results are indicative of the influence of temperature on the pest’s developmental stages and survival, they do not allow us to accurately measure these limits, as some field studies lack full control of all potentially influencing elements such as relative humidity and thermal variation. Daily variation in temperature can be very important and should be accurately recorded in situ when conducting field experiments on the thermal requirements of insects [22]. Furthermore, rigorous laboratory approaches using current technological resources are needed for precise knowledge of the limiting factors that regulate the populations of *T. erytreae.* So far, to our knowledge, the only published study on the effect of constant temperatures on the life cycle of *T. erytreae* was carried out by Aidoo et al. (2022) [23] but without investigating the influence of relative humidity.

As the pest continues its spread in the Iberian Peninsula and approaches citrus-producing regions, the objective of this study is to provide new information about the influence of different temperature and relative humidity regimes (mean and extreme) on the development, fecundity, and survival of *T. erytreae*. This data will help predict the real risk of *T. erytreae* and its potential geographical distribution and establishment in still uncolonized areas of the Iberian Peninsula and other citrus-producing countries in the Mediterranean. Furthermore, the information provided will help to predict the geographical distribution of the African citrus psyllid under a climate warming scenario.

## 2. Materials and Methods

This research was carried out in laboratories of two research centers in the Iberian Peninsula: The Phytopathological Station Areeiro (EFA, Deputación de Pontevedra, Spain) and the Vairão Agrarian Campus (Universidade do Porto, Porto, Portugal).

### 2.1. Mass Rearing of T. erytreae

In both centers, mass rearing was initiated by collecting adults from infested *Citrus limon* (Eureka cultivar) trees from Spain and Portugal. *Citrus limon* plants (cv. Eureka, 1–2 years old) at phenological stage 15 on the BBCH scale, i.e., “more leaves visible, not yet at full size” [24], were used for rearing. A total of 600 female and male adults (sex ratio of 1:1) were introduced into insect-rearing cages to start the colony. *Citrus limon* was selected for its ability to produce young shoots throughout the year [25] and is the most widely distributed citrus species in the area of study.

Insect rearing was carried out inside environmentally controlled chambers (Aralab^®^, SN1359, Rio de Mouro, Portugal) set at a temperature of 25 °C ± 2 °C, a relative humidity of 70 ± 2%, and a 14:10 h (D:N) photoperiod. At EFA, an additional breeding facility was established in a glasshouse with a double anti-thrips mesh door and equipped with an air conditioner device. The aim of this facility was to reinforce rearing facilities. and to have a backup in case of incidental issues (i.e., entry of the parasitoid *T. dryi*). In all cases, the plants inside the cages were replaced when the sprouting was inadequate and/or when adults emerged.

To guarantee continuous plant growth, periodic pruning was carried out, and fertigation was performed every 20 days with NPK Ultrasol 18.18.18 +2MgO (SQM Iberian, Barcelona, Spain). Micronutrient solution (Welgro Micromix, C.Q. Massó, Barcelona, Spain) was added occasionally.

### 2.2. Development, Fecundity, Fertility, Mortality, and Longevity of T. erytreae at Different Constant Temperatures and Relative Humidities

The influence of temperature on the duration of each developmental stage was studied at 10 °C, 15 °C, 20 °C, 25 °C, and 30 °C ± 1 °C, maintaining a constant relative humidity (70% ± 2%) and photoperiod (14:10 h D:N). To determine the influence of RH, temperature (25 °C ± 1 °C) and photoperiod (14:10 h D:N) were fixed, and relative humidity was modified to 40%, 70%, and 90% (±2%). The studies were carried out in environmentally controlled growth chambers (Aralab^®^ S600). *Citrus limon* plants (cv. Eureka) at the 15th phenological stage of the BBCH scale ([23] Agustí et al., 1995) were used. To conduct the study on the influence of temperature on insect development, a single plant was introduced in a cage covered by a polyester mesh and infested with 15 couples (sex ratio 1:1) of *T. erytreae*. Insects were kept inside cages for 7 days to ensure oviposition. Then, all adults were removed by aspiration and the plants were removed from the cages to facilitate daily evaluations of nymphal instars. Evaluation of the developmental time of the immature stages was carried out using a portable microscope (Pocket Digital Microscope, Levenhuk DTX 700 Mobi) until the emergence of adults. If adults did not emerge within 30 days after the 1st nymphal stage was detected, that replicate was rated as “development not completed”. The longevity of both female and male adults was also recorded. The number of replicates varied between 9 and 23 according to the experiment.

In addition to the developmental period, fecundity and fertility were studied at various temperatures and relative humidities by confining a single young couple of *T. erytreae* in lemon plants inside single cages. All studies were conducted following the same methodology as previously described, using 1-year-old Eureka lemon plants at the 15th growth stage. The number of eggs laid and emerged nymphs and adults per plant were evaluated daily until adult emergence. The number of valid replicates (number of couples) was 12 to 14.

### 2.3. Effect of Extreme Temperatures on Egg Hatching and Adult Mortality of T. erytreae

Lemon plants at the 12–13 BBCH growth stage (about 30 cm average height) placed in insect cages (built with plastic and polyester mesh to facilitate aeration) were used for experiments. All experiments were conducted in a Fitoclima Aralab^®^ (S600) (Rio de Mouro, Portugal) growth chamber, with a fixed photoperiod of 14:10 h (D:N) and relative humidity of 70 ± 10%. The purpose of this study was to find out the temperature thresholds in which embryonic development collapsed.

The study was started with a fixed temperature of 28 °C and was replicated with gradually lower and higher temperatures (at 5 °C temperature intervals below and above 28 °C) until reaching a temperature at which eggs would not hatch (after a 20-day incubation period). Thus, egg hatching was finally monitored at constant temperatures of 33 °C, 28 °C, 23 °C, 18 °C, 13 °C, and 8 °C, with a constant RH of 70% and a 14:10 h (D:N) photoperiod. For that purpose, 10 newly sprouted *Citrus limon* plants (10 replicates) were infested with a single young couple of *T. erytreae* and kept in the insectary at a temperature of 23 ± 5 °C, a relative humidity of 70 ± 10%, and a photoperiod of 14:10 (D:N). After egg laying and when at least 90 eggs/plant were observed, adults were removed from the cage and all plants were placed in the growth chamber at each given temperature. Egg development was monitored daily in each plant to record the days elapsed since egg laying until the first 5 and 50 neonate nymphs were observed.

In a parallel experiment, adult mortality was determined at temperatures of 3 °C, 6 °C, 15 °C, 20 °C, 25 °C, and 34 °C, maintaining a constant RH (70%) and a 14:10 h (D:N) photoperiod. For this purpose, five young *Citrus limon* plants (replicates) were introduced in a cage and infested with 18 young adults of *T. erytreae* (1:1 sex ratio, 7 days old) per cage. The soil below the plants was covered with white paper for easier visualization and collection of dead adults. Adult mortality was determined by counting the number of dead adults daily at different exposure times (24 h, 48 h, 72 h) and each tested temperature.

### 2.4. Statistical Analysis

Development, fecundity, fertility, mortality, and longevity data of *T. erytreae* at varying constant temperatures and relative humidities were transformed and then checked for normality by a Shapiro–Wilk test (*p* < 0.05) to decrease heteroscedasticity and achieve a normal distribution. Then, the transformed data were analyzed using a one-way ANOVA followed by a pairwise least significant difference (LSD) (*p* < 0.05) test using Statview 4.0 software for Macintosh [26].

To evaluate the effect of temperature on egg hatching and adult mortality, a Kruskal–Wallis H test (*p* < 0.05) followed by a Dunn Post Hoc test (adjusted by Bonferroni correction) was used to compare the influence of different temperatures on embryogenic development and mortality rate.

## 3. Results

### 3.1. Life Table Statistics of T. erytreae at Different Temperatures

*Trioza erytreae* completed its whole life cycle at 15, 20, and 25 °C and a constant RH of 70%. However, the life cycle was not completed at a constant temperature of 10 and 30 °C (Table 1) as the preoviposition period was delayed and nymphs were unable to reach the adult stage. Results showed that the duration of the life cycle is inversely related to temperature increase. The shortest life cycle for the species occurred at 25 °C, with an average duration of 23.9 days, increasing to 29.7 and 46.7 days at 20 and 15 °C, respectively. This increase in the duration of the life cycle is due to the extension of all phases of development (Table 1), with significant differences (*p* < 0.01) between the elapsed developmental time at each temperature. The effect of temperature on the duration of the cycle was more pronounced at the preoviposition stage, as it practically doubled the time required for oviposition at 20 °C versus 25 °C. The same trend was observed when the preoviposition period at 15 and 20 °C was compared. The differences were less pronounced between two consecutive temperatures for the egg incubation and nymphal developmental time.

Adult longevity and nymphal mortality rate were also evaluated (Table 2). Regarding longevity, females lived longer than males in all cases. Moreover, a decrease in temperature resulted in an increase in the lifespan of both males and females, with significant differences between consecutive temperatures. The increase in longevity was more evident between 20 °C and 15 °C, and the value obtained at this temperature doubled the one recorded at 25 °C.

The percentage of nymphal mortality, calculated by the difference between the number of nymphs and the number of adults who emerged, was not significantly (*p* > 0.05) different among the three temperatures tested, although the mean mortality was twice as much at 25 than at 20 °C. Despite the fact that the mortality rate at 20 °C and 15 °C was almost half that at 25 °C, the values obtained were not statistically different.

Temperatures of 10 and 30 °C were detrimental to the development of *T. erytreae*. At 10 °C (n = 9), oviposition occurred only in 50% of the plants where adults were released, and the preoviposition period was prolonged for more than 20 days, compared to 3.4 days at 25 °C. Nymph emergence was observed in only one repetition and it was delayed for more than 30 days at 10 °C. If no nymph hatching was recorded after 30 days, visual observations were stopped.

Then, all plants with unhatched eggs previously maintained at 10 °C were transferred to a growth chamber with optimal development conditions (25 ± 1 °C and 70 ± 2%). Nymphs were born after the seventh day and developed into adults, completing their life cycle in a similar time period as those that were maintained at 25 °C from the beginning of the egg stage. Thus, unhatched eggs maintained for 30 days at 10 °C were able to hatch when the temperature was raised to 25 °C.

At 30 °C (n = 9), adults died in a short time period (after 3.3 days for males and after 6.6 days for females), and some adults did not even change their color from light green (immediately after emergence) to brown (mature adult). The mean preoviposition period was 3.8 days, with extreme values of 2 and 7 days. Egg hatching only occurred in 33% of the plants eight days after oviposition at a similar rate to that at 20 °C. Nymph emergence was very limited at 30 °C and none of them devolved beyond the N2 instar. Unhatched eggs were examined under a binocular microscope and the majority had undergone complete invagination.

The effect of temperature on the reproduction parameters of *T. erytreae* is shown in Table 2. The highest fecundity was recorded at 25 °C, with an average mean value of 208.3 eggs per female, which was significantly (*p* < 0.01) higher than at the other two temperatures. On the other hand, the lowest percentage of hatched eggs (fertility) was recorded at 25 °C but no significant (*p* = 0.56) differences were found in fertility between the temperatures tested. In fact, the fertility rate was inversely related to temperature increase.

### 3.2. Influence of the Relative Humidity in the Life Cycle of T. erytreae

The preoviposition period was significantly (*p* < 0.01) shorter at an RH of 70% (3.4 days) than at 40% (4.5 days) and 90% (4.4 days) (Table 3). The egg incubation period was significantly longer at 70% (7.2 days) than at 40% (6.3 days) and at 90% (6.6 days) with lower statistical significance for this value. The entire life cycle of the insect was fully completed only when the RH was set to 70%. At 40% and 90%, nymphal development did not progress beyond the N3 instar, at least after a 30-day observation period since the eggs hatched (which was established as the test limit).

Relative humidity also influenced the longevity of adults, which was longest at 70%, being significantly lower for females under an RH of 40% (Table 4). Humidity also had a strong impact on nymphal survival. The values of nymphal mortality were similar at 70 and 90% RH but significantly higher at 40% (mortality raised up to 68%). In addition, all nymphs showed clear signs of dehydration when RH dropped to 40%.

Relative humidity also influenced the reproduction of *T. erytreae*. The highest fecundity was obtained at 70%, with an average of 208 eggs per female, which was double that at 90% (112) and 40% (94). Fertility was also highest at 70% (60.7%), although it was very similar and therefore not significantly different from that at 90% (59%). However, low humidity (RH = 40%) significantly reduced the fertility of *T. erytreae* (it dropped to 32%) (Table 4).

### 3.3. Effect of Extreme Temperatures on Egg Hatching and Adult Mortality of T. erytreae

The study conducted on the time required for egg hatching (incubation period) at extreme temperatures revealed that eggs failed to hatch under constant temperatures of 8 °C and 33 °C after a 25-day observation period (Table 5).

Similarly, it was observed that adult mortality was highest at a constant temperature of 34 °C (over 50% after a 3-day exposure to that temperature). These studies also showed that mortality was highest for both higher and lower temperature extremes (Figure 1).

## 4. Discussion

At the present time, Mediterranean citriculture is challenged by both psyllid vector species of HLB, with *T. erytreae* expanding from the West (Iberian Peninsula) and *D. citri* expanding from the East (Israel and Cyprus). Thus, precise knowledge about the life cycle of *Trioza erytreae* and its limiting environmental conditions is a key element for predicting its geographical distribution and designing effective and novel control tactics against this key pest. In other words, the invasive nature of this pest, in association with climate change, makes the rigorous knowledge of the impact of climatic factors on its life cycle an important step in predicting and preventing its introduction and establishment in new regions. However, detailed information on the duration of the life cycle of *T. erytreae* and the limiting temperature and relative humidity for its development and survival was not well known. Our study provides key information on the response of *T. erytreae* to varying temperatures and humidities that could be used to model the geographical distribution and the risk of spread of *T. erytreae* across Europe and elsewhere. More precisely, our work provides new insights into the environmental factors limiting the populations of the African citrus psyllid. We found that temperatures above 33 °C prevent egg hatching and can increase adult mortality by up to 50%. Furthermore, an RH around 40% is very detrimental to nymphal survival (mortality reaching almost 70%).

According to our study, the duration of the total developmental time from egg to adult ranged from 23.9 days at 25 °C to 46.7 days at 15 °C (Table 1). Our results were quite similar in the range of 18–24 °C but differed at 15 °C to those reported by Aidoo et al. (2022) [23]. They reported a total developmental time (from the egg to the fifth instar nymph stage) of 56.2 days at 15 °C, much longer than the values obtained in our study (46.7 days). Such discrepancies could be related to the way that both studies were conducted or to genetic differences between the two *T. erytreae* populations. In our study, we did not manipulate the nymphs nor eggs in any way, while Aidoo et al. (2022) [23] transferred the emerged nymphs to other shoots, which could explain why the duration of the life cycle was extended up to 56 days at 15 °C. Also, the preoviposition time was not reported by Aidoo et al. (2022) [23], which we found to be longer with decreasing temperatures. In our study, preoviposition lasted an average of 3.4 days at 25 °C, 5.6 days at 20 °C, and 10.5 days at 15 °C. These values are similar to those reported in the literature [18], except for the one reported by Catling (1973) [20] at 14–16 °C, which was 6–7 days in an outdoor insectary and therefore without full temperature control.

Catling (1973) [20] also noted that under variable temperature conditions, the duration of the egg incubation period was inversely proportional to the recorded mean temperature: the eggs developed in 6 days at an average temperature of 25 °C and 15 days at 14 °C. Aidoo et al. (2022) [23] reported an incubation period of 5.3 days at 24 °C. Such values are consistent with those of our study, where we found that the shortest egg incubation period for nymph emergence was 6 days at 23 °C and the longest at 13 °C (15.6 days). The temperature of 13 °C stands out from the others with a longer egg developmental time [20,23]. However, our results showed that when the temperature increased to 28 °C, the incubation period was extended to 8.4 days, suggesting that higher temperatures can also extend embryonic development. In fact, no eggs hatched at a temperature of 33 °C. Thus, the incubation period increased with decreasing temperatures but only at values below 23 °C. When analyzing the incubation period until the hatching of 50 nymphs, the temperature of 23 °C showed significant differences when compared to the other temperatures tested, thus confirming that such a temperature is the closest to optimal for the embryonic development of *T. erytreae*, with the lowest number of days until hatching. At 33 and 8 °C, there was no hatching at all within the maximum time of observation, which was limited to 25 days (Table 5). Thus, 33 and 8 °C should be considered the upper and lower temperature thresholds for the development of *T. erytreae* eggs.

Regarding the nymphal developmental time, our findings were also consistent with those reported by Catling (1973) [20], who found that nymphs developed in 17 days at 25 °C and extended to 43 days at 14 °C, indicating that daily average temperatures of 10 to 12 °C are the lower temperature limit for the development of immature stages. Likewise, in our study, no nymphs evolved into adults at 10 °C (Table 1), and the same happened in the Aidoo et al. (2022) [23] study.

Our study showed that temperatures below 10 °C and above 30 °C were the limits for *T. erytreae* development. The upper limit is consistent with the findings of Moran and Blowers (1967) [18], who reported that oviposition, hatching, and survival were drastically reduced at 32 °C. This is also supported by laboratory research conducted by Aidoo et al. (2022) [23], who found that nymphs were unable to complete their development at 10 °C and 27 °C. Furthermore, the preoviposition period was slowed down at 10 °C, according to Moran and Blowers (1967) [18].

Our results clearly showed that 25 °C was the most suitable temperature for the development (Table 1) and reproduction (Table 2) of *T. erytreae* according to what has been discussed so far. On the other hand, nymphal mortality was highest at 25 °C (40.2%), although without significant differences compared to the lowest mortality (20.8%), which was recorded at 20 °C (Table 2). Our data are similar to those previously reported by Aidoo et al. (2022) [23], who obtained a nymphal mortality of 46.1% at 24 °C. These relatively high mortality rates are lower than those reported by other authors (60–70%) under natural or controlled conditions but similar to those considered natural in certain areas of South Africa ([21,27]).

In our work, we found that adult longevity was higher at lower temperatures, with the lifespan of both males and females almost doubling at 15 °C compared to 25 °C. The data we obtained fall within the range reported by Moran and Blowers (1967) [18] under different environmental conditions. However, we found that the longevity of females was higher than males (Table 2), which has also been reported for insects belonging to other families of Psylloidea [28].

Our study showed that the optimal temperature for adult survival is in the range of 15–25 °C, with the minimum mortality being at 25 °C. On the other hand, the maximum mortality was recorded for a temperature of 34 °C (Figure 1). Adult mortality at 34 °C was significantly different from the other tested temperatures, showing much higher mortality rates after a short period of exposure (24 h), thus extremely high temperatures cause greater mortality in adults than low temperatures. Catling (1969) [29] showed that eggs and the first instar were the most affected by extremely high temperatures that led to an accentuated decrease in the population of *T. erytreae*, and only part of the adults and advanced instar nymphs survived ([29,30]). However, we found that adult mortality at 3 and 34 °C did not differ from each other after 48 h of exposure. For the 72 h exposure time, adult mortality at temperatures of 3, 6, and 34 °C was significantly different from the mortality obtained at the other tested temperatures, confirming that after a long exposure period, extremely high temperatures (34 °C), but also extremely low temperatures (3 °C), may cause high mortality in adults.

One of the novel aspects of our study is that we were able to determine the response of *T. erytreae* to high and low relative humidities (Table 3 and Table 4). At 40% and 90% RH, adult emergence did not occur after 30 days, and the highest nymphal mortality (68.4%) was obtained at low RH (40%). Our data support field experiments that showed how hot and dry days with high water vapor pressure deficits (VPDs) were lethal to *T. erytreae* eggs and first instar nymphs [29]. Green and Catling (1972) [19] related VPD to the survival of *T. erytreae* early life stages and found that no survival was observed when the VPD was above 56 mbar. Thus, regions in Portugal with high VPD values such as Castelo Branco and Portalegre (Center), Beja (Alentejo), Alte, and Norinha (Algarve) are not suitable for the establishment of *T. erytreae* ([31]). Similarly, citrus regions in southwestern Spain (Huelva) with a hot and relatively dry climate similar to Algarve would be also unsuitable for the establishment of *T. erytreae*. Also, our study showed that fecundity was significantly reduced at 90 and 40% RH, and fertility was lowest at 40% RH. Following these lines, a low RH of 40% was the most detrimental humidity among all tested for survival, development, and reproduction (Table 2). Thus, in summary, we can conclude that the African citrus psyllid prefers to develop at a relative humidity of around 70%, 40% being the most detrimental. Furthermore, *T. erytreae* seems not well adapted to dry and hot regions, where the relative humidity is around or below 40% and temperatures can reach values above 34 °C, which will kill more than 50% of the adult population. Moreover, its establishment and spread will be maximum in regions with a temperate and humid climate, being rare in regions where dry and hot weather conditions predominate. It is important to highlight here that the mechanisms leading to positive or negative impacts of extremely high temperatures on insects can only be resolved from integrative approaches considering fluctuating thermal regimes. Thus, thermal extremes, probably more than the gradual increase in mean temperature, drive insect responses to climate change [32]. Thus, we can conclude that under a climate warming scenario, *T. erytreae* will suffer in high temperatures and dry conditions, suggesting that it will be displaced to geographical regions with less extreme summer conditions at higher latitudes.

## Figures and Tables

**Figure 1 insects-15-00062-f001:**
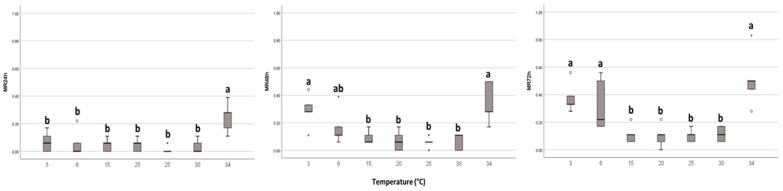
Mortality rate (MR) of adults of *T. erytreae* at different temperatures after 24 h, 48 h, and 72 h of exposure. Box-plot followed by the same letter for each temperature did not differ according to Dunn test (*p* < 0.05). The hollow circles and asterisks represent outliers, as “out” values and “far out” values, respectively.

**Table 1 insects-15-00062-t001:** Life table statistics (mean ± SE) * of *T. erytreae* at different temperatures (RH = 70 ± 2%) (numerical values represent days).

Temperature (°C)	Preoviposition	Incubation	Nymphal Development	Total Cycle (from Egg to Adult)
10 ^†^ (n = 9)	-	-	-	-
15 (n = 12)	10.5 ± 0.5 a	13.5 ± 0.4 a	33.4 ± 0.5 a	46.7 ± 0.6 a
20 (n = 18)	5.5 ± 0.2 b	8.3 ± 0.2 b	21.3 ± 0.3 b	29.7 ± 0.4 b
25 (n = 18)	3.4 ± 0.1 c	7.2 ± 0.2 c	16.4 ± 0.2 c	23.9 ± 0.2 c
30 ^‡^ (n = 9)	-	-	-	-

* Means within columns followed by different letters are significantly different according to a one-way ANOVA followed by an LSD test (*p* < 0.05). ^†^ Preoviposition was delayed and no nymphs turned into adults. ^‡^ Some nymphs emerged but none evolved into adults.

**Table 2 insects-15-00062-t002:** Mean ± SE * longevity of males and females (days), nymphal mortality (%), fecundity (no. of eggs/female), and fertility (%) of *T. erytreae* at different temperatures (RH = 70 ± 2%).

	TEMPERATURE (°C)				
	10 ^†^	15	20	25	30 ^‡^
Longevity ♂	-	36.8 ± 1.2 a	22.4 ± 1.8 b	17.2 ± 0.9 c	-
Longevity ♀	-	44.2 ± 2.6 a	28.4 ± 1.1 b	22.8 ± 0.7 c	-
Nymphal mortality (%)	-	25.8 ± 7.0 a	20.8 ± 4.6 a	40.2 ± 6.3 a	-
Fecundity/♀	-	141.2 ± 8.03 a	126.7 ± 11.2 a	208.3 ± 17.5 b	-
Fertility (%)	-	70.7 ± 6.5 a	68.4 ± 7.2 a	60.7 ± 6.7 a	-

* Means within lines followed by different letters are significantly different according to a one-way ANOVA followed by an LSD test (*p* < 0.05). Number of replicates was 12. ^†^ Preoviposition was delayed and no nymphs turned into adults. ^‡^ Some nymphs emerged but none evolved into adults.

**Table 3 insects-15-00062-t003:** Life table statistics (mean + SE) * of *T. erytreae* at a relative humidity of 40, 70, and 90% and a constant temperature of 25 ± 1 °C (time in days).

Relative Humidity (%)	Preoviposition	Incubation	Nymphal Development	Total Cycle (from Egg-Adult)
40 ^†^ (n = 18)	4.5 ± 0.3 a	6.3 ± 0.3 a	-	-
70 (n = 18)	3.4 ± 0.12 b	7.2 ± 0.22 b	16.4 ± 0.23	23.9 ± 0.18
90 ^†^ (n = 23)	4.4 ± 0.27 a	6.6 ± 0.15 ab	-	-

* Means within columns followed by different letters are significantly different according to a one-way ANOVA followed by an LSD test (*p* < 0.05). ^†^ No nymphs developed into adults after 30 days of observation.

**Table 4 insects-15-00062-t004:** Mean ± SE * longevity of males and females (in days), nymphal mortality (%), fecundity (no. of eggs/female), and fertility (%) of *T. erytreae* at a relative humidity of 40, 70, and 90% (T = 25 ± 1 °C).

	RELATIVE HUMIDITY (%)		
	40	70	90
Longevity ♂	16.2 ± 0.7 a	17.2 ± 0.9 a	16.1 ± 1.2 a
Longevity ♀	18.1 ± 0.7 b	22.8 ± 0.7 a	21.0 ± 1.4 a
Nymphal mortality (%)	68.4 ± 4.7 b	40.2 ± 6.3 a	53.1 ± 2.5 a
Fecundity	93.9 ± 9.7 a	208.2 ± 17.5 b	112.5 ± 8.3 a
Fertility (%)	32.1 ± 6.5 b	60.7 ± 6.7 a	59 ± 4.3 a

* Means within lines followed by different letters are significantly different according to a one-way ANOVA followed by an LSD test (*p* < 0.05). The number of replicates was 12.

**Table 5 insects-15-00062-t005:** Average duration of incubation period of *T. erytreae* eggs until hatching of 5 and 50 nymphs at different temperatures (8, 13, 18, 23, 28, and 33 °C).

Temperature (°C)	5 Nymphs	50 Nymphs
8 ^†^	-	-
13	15.6 ± 0.6 a	22.0 ± 0.6 a
18	7.8 ± 0.6 bc	13.2 ± 0.8 b
23	6.2 ± 0.3 c	8.2 ± 0.2 c
28	8.4 ± 0.3 b	13.1 ± 0.5 b
33 ^†^	-	-

Means within columns followed by different letters are significantly different according to LSD test (*p* < 0.05) to Kruskal–Wallis test followed by Dunn test (*p* < 0.05). ^†^ Eggs failed to hatch after a 25-day observation period.

## Data Availability

The data presented in this study and used for statistical analysis are available on request from the corresponding author.

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
