# Peer review of "The African Psyllid Trioza erytreae Del Guercio (1918) Is Very Sensitive to Low Relative Humidity and High Temperatures"

_insects, 2024, doi:10.3390/insects15010062_

Round 1

Reviewer 1 Report

Comments and Suggestions for Authors

This study investigated the mortality, development and reproduction of the African psyllid, a vector of Huanglongbing in citrus. The data presented in this paper can help predict the geographical distribution of T. erytreae and also help assess risk. This paper is well written and the experiments were well designed. I  have a few minor suggestions.

In Tables 1 and 2, please change "HR" to "RH" for consistency with the standard abbreviation for Relative Humidity.

It would be helpful to specify in Table 1 that the numerical values represent days.

Author Response

All changes suggested by reviewer have been made

Reviewer 2 Report

Comments and Suggestions for Authors

The authors investigated the influence of temperature and relative air humidity extremes on survival and reproduction of the African citrus psyllid, Trioza erytreae, an invasive pest and the vector of bacterium causing disease of citrus trees. With this aim they conducted simple laboratory experiments and demonstrated that temperature of 30°C and higher as well as relative air humidity of 40% and lower are not suitable for the development of this insect. Generally speaking, these or similar results can be expected and therefore the present study is not of high fundamental novelty and value but the exact determination of the limits of the optimum and tolerance zones is very important for the prediction of the limits of the further invasion. The experiments were well planned and conducted, the data were correctly analyzed. The text is clearly written. The figures are easily understandable. Thus, the manuscript can be published although it needs a number of corrections and improvements and, in particular, the authors have to justify the method used (see below).

My single (but very important) concern about the present study is that the authors used only constant conditions, although in nature both temperature and air humidity are subjected to daily rhythms. In this context, I would not agree that the results of field studies are less informative than the results of laboratory experiments conducted under controlled constant conditions (lines 100-102). Yes, indeed, filed studies lack full control of all environmental factors, but these factors (including thermal variation – see line 102) still can be at least recorded.

In the present study, only constant temperatures were used whereas natural daily variations in temperature combined with microclimate selection can allow insects to survive even in regions with extreme environmental conditions. Mean temperatures provided by usual meteorological stations are more or less OK for the prediction of phenology (as the rate of development linearly depends on temperature within the optimal zone) but not for the prediction of survival rate. This in no way means that laboratory experiments conducted under various combinations of constant temperature and air humidity are useless, but the authors should consider this aspect of the study both in the Introduction (to justify methods) and in the Discussion (to note the limits of the practical application of their results).

Recently published relevant papers on this topic could be cited, for example:

Käfer, H., Kovac, H., & Stabentheiner, A. (2023). Habitat Temperatures of the Red Firebug, Pyrrhocoris apterus: The Value of Small-Scale Climate Data Measurement. Insects, 14(11), 843 (about microclimate)

Ma, C. S., Ma, G., & Pincebourde, S. (2021). Survive a warming climate: insect responses to extreme high temperatures. Annual Review of Entomology, 66, 163-184 (about the importance of daily rhythms of temperature).

Minor corrections and comments

Lines 5 and 9: Please, include country (Spain) in the full postal address.

Lines 42, 46 etc. Delete names of the authors and years of publication, only the numbers of the references in the list should be given.

Table 1: Why the numbers of replicates twice differ between temperatures?

Line 239: Degree sign (the 11th symbol in this line) should not be underlined.

Lines 256, 263: see my comment to line 239 and please check the whole text.

Line 265: I guess that superscript asterisk after SE means footnote (as in Table 1) but none of the footnotes to this table is labeled by the asterisk.

Tables 2 and 5 – I guess that asterisks in these tables mean the absence of data but this should be clearly explained. Besides, I would recommend using dashes instead asterisks to avoid confusion with asterisks in the footnote.

Lines 285 and 297: I think that all footnotes should be labeled as in Table 1 (see line 226).

Lines 387-393: Usually these final conclusions (“In summary, we found that...”) are placed at the end of the text.

Author Response

Thanks very much for the comments made by the reviewer.

Below you can find my answers:

"My single (but very important) concern about the present study is that the authors used only constant conditions, although in nature both temperature and air humidity are subjected to daily rhythms. In this context, I would not agree that the results of field studies are less informative than the results of laboratory experiments conducted under controlled constant conditions (lines 100-102). Yes, indeed, filed studies lack full control of all environmental factors, but these factors (including thermal variation – see line 102) still can be at least recorded."

Yes, we totally agree that daily rhythms are extremely important for the life cycle of insects such as the one used in our study. Including daily variations of temperature (for example using different day/night temperatures) would provide a more accurate outcome of our study. However, we decided to use constant temperatures because the main aim of our study was to find the upper and lower thresholds for development and survival. 

I have made changes as requested in lines 100-102 to highlight the importance of considering daily variations and accurate recording of such changes when conducting field studies. The use of thermal sensors located in the same place where the insect develops can provide very useful information, for example when constructing degree-day models.

I have also added in the introduction and discussion some new sentences and citations of some recent publications regarding climate data measurement (as requested by the reveiwer)

Minor corrections and comments have also been addressed and changes added to the manuscript

The number of replicates in Table 1 differ among temperatures because there was high mortality under extreme temperatures and the life cycle was not completed.

Reviewer 3 Report

Comments and Suggestions for Authors

I commend the authors for a well-designed and carefully conducted study despite all the experimental challenges. The manuscript can be reduced by at least 20% if English is properly used (see below) and the authors properly format data presentation. I often see the use of 4 to 5 significant figures in the presentation of data. That type of precision is often unnecessary particularly in the present manuscript. For example, what value does a presentation of P = 0.00001 over a P < 0.001, or F = 1.2534 instead of 1.25, or a mean value of 10.24 vs 10.2? I encourage the authors to trim their significant figures to 3 (i.e., 0.00).

The authors should also pay attention to when to spell out the full organism scientific name. For example, Trioza erytreae should be presented as T. erytreae after first occurrence and as Trioza erytreae at beginning of sentence. 

Comments on the Quality of English Language

I suggest that the authors seek assistance from a native English speaker/writer to improve the quality of the manuscript's English. I noticed that the authors often use informal style in some sentences. Classically, this is not desirable in scientific writing, but this point widely debated and depends on the journal's guidelines. Sometimes informal sentences tend to convey the information in a simpler form for readers to understand. 

Author Response

As requested by the reviewer the manuscript has been reviewed by a native English speaker and several changes have been made to improve reading and reducing the length of the manuscript. Informal style has been corrected and eliminated when necessary.

Also, the data presented has been trimmed to 2 significant figures

The name of Trioza erytreae has been shortened to T. erytreae where necessary

Round 2

Reviewer 2 Report

Comments and Suggestions for Authors

The authors addressed all my comments and I think that the present version of the manuscript can be published.